# Variational Policy Search via Trajectory Optimization

**Sergey Levine**
Stanford University
svlevine@cs.stanford.edu

**Vladlen Koltun**
Stanford University and Adobe Research
vladlen@cs.stanford.edu

## Abstract

In order to learn effective control policies for dynamical systems, policy search methods must be able to discover successful executions of the desired task. While random exploration can work well in simple domains, complex and high-dimensional tasks present a serious challenge, particularly when combined with high-dimensional policies that make parameter-space exploration infeasible. We present a method that uses trajectory optimization as a powerful exploration strategy that guides the policy search. A variational decomposition of a maximum likelihood policy objective allows us to use standard trajectory optimization algorithms such as differential dynamic programming, interleaved with standard supervised learning for the policy itself. We demonstrate that the resulting algorithm can outperform prior methods on two challenging locomotion tasks.

## 1 Introduction

Direct policy search methods have the potential to scale gracefully to complex, high-dimensional control tasks [12]. However, their effectiveness depends on discovering successful executions of the desired task, usually through random exploration. As the dimensionality and complexity of a task increases, random exploration can prove inadequate, resulting in poor local optima. We propose to decouple policy optimization from exploration by using a variational decomposition of a maximum likelihood policy objective. In our method, exploration is performed by a model-based trajectory optimization algorithm that is not constrained by the policy parameterization, but attempts to minimize both the cost and the deviation from the current policy, while the policy is simply optimized to match the resulting trajectory distribution. Since direct model-based trajectory optimization is usually much easier than policy search, this method can discover low cost regions much more easily. Intuitively, the trajectory optimization "guides" the policy search toward regions of low cost.

The trajectory optimization can be performed by a variant of the differential dynamic programming algorithm [4], and the policy is optimized with respect to a standard maximum likelihood objective. We show that this alternating optimization maximizes a well-defined policy objective, and demonstrate experimentally that it can learn complex tasks in high-dimensional domains that are infeasible for methods that rely on random exploration. Our evaluation shows that the proposed algorithm produces good results on two challenging locomotion problems, outperforming prior methods.

## 2 Preliminaries

In standard policy search, we seek to find a distribution over actions $\mathbf{u}_t$ in each state $\mathbf{x}_t$, denoted $\pi_\theta(\mathbf{u}_t|\mathbf{x}_t)$, so as to minimize the sum of expected costs $E[c(\zeta)] = E[\sum_{t=1}^{T} c(\mathbf{x}_t, \mathbf{u}_t)]$, where $\zeta$ is a sequence of states and actions. The expectation is taken with respect to the system dynamics $p(\mathbf{x}_{t+1}|\mathbf{x}_t, \mathbf{u}_t)$ and the policy $\pi_\theta(\mathbf{u}_t|\mathbf{x}_t)$, which is typically parameterized by a vector $\theta$.

An alternative to this standard formulation is to convert the task into an inference problem, by introducing a binary random variable $\mathcal{O}_t$ at each time step that serves as the indicator for "optimality."

We follow prior work and define the probability of $\mathcal{O}_t$ as $p(\mathcal{O}_t = 1|\mathbf{x}_t, \mathbf{u}_t) \propto \exp(-c(\mathbf{x}_t, \mathbf{u}_t))$ [19]. Using the dynamics distribution $p(\mathbf{x}_{t+1}|\mathbf{x}_t, \mathbf{u}_t)$ and the policy $\pi_\theta(\mathbf{u}_t|\mathbf{x}_t)$, we can define a dynamic Bayesian network that relates states, actions, and the optimality indicator. By setting $\mathcal{O}_t = 1$ at all time steps and learning the maximum likelihood values for $\theta$, we can perform policy optimization [20]. The corresponding optimization problem has the objective

$$p(\mathcal{O}|\theta) = \int p(\mathcal{O}|\zeta)p(\zeta|\theta)d\zeta \propto \int \exp\left(-\sum_{t=1}^{T} c(\mathbf{x}_t, \mathbf{u}_t)\right)p(\mathbf{x}_1)\prod_{t=1}^{T}\pi_\theta(\mathbf{u}_t|\mathbf{x}_t)p(\mathbf{x}_{t+1}|\mathbf{x}_t, \mathbf{u}_t)d\zeta. \quad (1)$$

Although this objective differs from the classical minimum average cost objective, previous work showed that it is nonetheless useful for policy optimization and planning [20, 19]. In Section 5, we discuss how this objective relates to the classical objective in more detail.

## 3  Variational Policy Search

Following prior work [11], we can decompose $\log p(\mathcal{O}|\theta)$ by using a variational distribution $q(\zeta)$:

$$\log p(\mathcal{O}|\theta) = \mathcal{L}(q, \theta) + D_{\text{KL}}(q(\zeta)\|p(\zeta|\mathcal{O}, \theta)),$$

where the variational lower bound $\mathcal{L}$ is given by

$$\mathcal{L}(q, \theta) = \int q(\zeta) \log \frac{p(\mathcal{O}|\zeta)p(\zeta|\theta)}{q(\zeta)} d\zeta,$$

and the second term is the Kullback-Leibler (KL) divergence

$$D_{\text{KL}}(q(\zeta)\|p(\zeta|\mathcal{O}, \theta)) = -\int q(\zeta) \log \frac{p(\zeta|\mathcal{O}, \theta)}{q(\zeta)} d\zeta = -\int q(\zeta) \log \frac{p(\mathcal{O}|\zeta)p(\zeta|\theta)}{q(\zeta)p(\mathcal{O}|\theta)} d\zeta. \quad (2)$$

We can then optimize the maximum likelihood objective in Equation 1 by iteratively minimizing the KL divergence with respect to $q(\zeta)$ and maximizing the bound $\mathcal{L}(q, \theta)$ with respect to $\theta$. This is the standard formulation for expectation maximization [9], and has been applied to policy optimization in previous work [8, 21, 3, 11]. However, prior policy optimization methods typically represent $q(\zeta)$ by sampling trajectories from the current policy $\pi_\theta(\mathbf{u}_t|\mathbf{x}_t)$ and reweighting them, for example by the exponential of their cost. While this can improve policies that already visit regions of low cost, it relies on random policy-driven exploration to discover those low cost regions. We propose instead to directly optimize $q(\zeta)$ to minimize both its expected cost and its divergence from the current policy $\pi_\theta(\mathbf{u}_t|\mathbf{x}_t)$ when a model of the dynamics is available. In the next section, we show that, for a Gaussian distribution $q(\zeta)$, the KL divergence in Equation 2 can be minimized by a variant of the differential dynamic programming (DDP) algorithm [4].

## 4  Trajectory Optimization

DDP is a trajectory optimization algorithm based on Newton's method [4]. We build off of a variant of DDP called iterative LQR, which linearizes the dynamics around the current trajectory, computes the optimal linear policy under linear-quadratic assumptions, executes this policy, and repeats the process around the new trajectory until convergence [17]. We show how this procedure can be used to minimize the KL divergence in Equation 2 when $q(\zeta)$ is a Gaussian distribution over trajectories. This derivation follows previous work [10], but is repeated here and expanded for completeness.

Iterative LQR is a dynamic programming algorithm that recursively computes the value function backwards through time. Because of the linear-quadratic assumptions, the value function is always quadratic, and the dynamics are Gaussian with the mean at $f(\mathbf{x}_t, \mathbf{u}_t)$ and noise $\epsilon$. Given a trajectory $(\bar{\mathbf{x}}_1, \bar{\mathbf{u}}_1), \ldots, (\bar{\mathbf{x}}_T, \bar{\mathbf{u}}_T)$ and defining $\hat{\mathbf{x}}_t = \mathbf{x}_t - \bar{\mathbf{x}}_t$ and $\hat{\mathbf{u}}_t = \mathbf{u}_t - \bar{\mathbf{u}}_t$, the dynamics and cost function are then approximated as following, with subscripts $\mathbf{x}$ and $\mathbf{u}$ denoting partial derivatives:

$$\hat{\mathbf{x}}_{t+1} \approx f_{\mathbf{x}t}\hat{\mathbf{x}}_t + f_{\mathbf{u}t}\hat{\mathbf{u}}_t + \epsilon$$

$$c(\mathbf{x}_t, \mathbf{u}_t) \approx \hat{\mathbf{x}}_t^{\mathrm{T}} c_{\mathbf{x}t} + \hat{\mathbf{u}}^{\mathrm{T}} c_{\mathbf{u}t} + \frac{1}{2}\hat{\mathbf{x}}_t^{\mathrm{T}} c_{\mathbf{xx}t}\hat{\mathbf{x}}_t + \frac{1}{2}\hat{\mathbf{u}}_t^{\mathrm{T}} c_{\mathbf{uu}t}\hat{\mathbf{u}}_t + \hat{\mathbf{u}}_t^{\mathrm{T}} c_{\mathbf{ux}t}\hat{\mathbf{x}}_t + c(\bar{\mathbf{x}}_t, \bar{\mathbf{u}}_t).$$

Under this approximation, we can recursively compute the Q-function as follows:

$$Q_{\mathbf{xx}t} = c_{\mathbf{xx}t} + f_{\mathbf{x}t}^{\mathrm{T}} V_{\mathbf{xx}t+1} f_{\mathbf{x}t} \qquad Q_{\mathbf{uu}t} = c_{\mathbf{uu}t} + f_{\mathbf{u}t}^{\mathrm{T}} V_{\mathbf{xx}t+1} f_{\mathbf{u}t} \qquad Q_{\mathbf{ux}t} = c_{\mathbf{ux}t} + f_{\mathbf{u}t}^{\mathrm{T}} V_{\mathbf{xx}t+1} f_{\mathbf{x}t}$$

$$Q_{\mathbf{x}t} = c_{\mathbf{x}t} + f_{\mathbf{x}t}^{\mathrm{T}} V_{\mathbf{x}t+1} \qquad\qquad Q_{\mathbf{u}t} = c_{\mathbf{u}t} + f_{\mathbf{u}t}^{\mathrm{T}} V_{\mathbf{x}t+1},$$

as well as the value function and linear policy terms:

$$V_{\mathbf{x}t} = Q_{\mathbf{x}t} - Q_{\mathbf{ux}t}^{\mathrm{T}} Q_{\mathbf{uu}t}^{-1} Q_{\mathbf{u}} \qquad \mathbf{k}_t = -Q_{\mathbf{uu}t}^{-1} Q_{\mathbf{u}t}$$

$$V_{\mathbf{xx}t} = Q_{\mathbf{xx}t} - Q_{\mathbf{ux}t}^{\mathrm{T}} Q_{\mathbf{uu}t}^{-1} Q_{\mathbf{ux}} \qquad \mathbf{K}_t = -Q_{\mathbf{uu}t}^{-1} Q_{\mathbf{ux}t}.$$

The deterministic optimal policy is then given by

$$g(\mathbf{x}_t) = \bar{\mathbf{u}}_t + \mathbf{k}_t + \mathbf{K}_t(\mathbf{x}_t - \bar{\mathbf{x}}_t).$$

By repeatedly computing the optimal policy around the current trajectory and updating $\bar{\mathbf{x}}_t$ and $\bar{\mathbf{u}}_t$ based on the new policy, iterative LQR converges to a locally optimal solution [17]. In order to use this algorithm to minimize the KL divergence in Equation 2, we introduce a modified cost function $\bar{c}(\mathbf{x}_t, \mathbf{u}_t) = c(\mathbf{x}_t, \mathbf{u}_t) - \log \pi_\theta(\mathbf{u}_t | \mathbf{x}_t)$. The optimal trajectory for this cost function approximately[1] minimizes the KL divergence when $q(\zeta)$ is a Dirac delta function, since

$$D_{\mathrm{KL}}(q(\zeta) \| p(\zeta | \mathcal{O}, \theta)) = \int q(\zeta) \left[ \sum_{t=1}^{T} c(\mathbf{x}_t, \mathbf{u}_t) - \log \pi_\theta(\mathbf{u}_t | \mathbf{x}_t) - \log p(\mathbf{x}_{t+1} | \mathbf{x}_t, \mathbf{u}_t) \right] d\zeta + \text{const.}$$

However, we can also obtain a Gaussian $q(\zeta)$ by using the framework of linearly solvable MDPs [16] and the closely related concept of maximum entropy control [23]. The optimal policy $\pi_{\mathcal{G}}$ under this framework minimizes an augmented cost function, given by

$$\tilde{c}(\mathbf{x}_t, \mathbf{u}_t) = \bar{c}(\mathbf{x}_t, \mathbf{u}_t) - \mathcal{H}(\pi_{\mathcal{G}}),$$

where $\mathcal{H}(\pi_{\mathcal{G}})$ is the entropy of a stochastic policy $\pi_{\mathcal{G}}(\mathbf{u}_t | \mathbf{x}_t)$, and $\bar{c}(\mathbf{x}_t, \mathbf{u}_t)$ includes $\log \pi_\theta(\mathbf{u}_t | \mathbf{x}_t)$ as above. Ziebart [23] showed that the optimal policy can be written as

$$\pi_{\mathcal{G}}(\mathbf{u}_t | \mathbf{x}_t) = \exp(-Q_t(\mathbf{x}_t, \mathbf{u}_t) + V_t(\mathbf{x}_t)),$$

where $V$ is a "softened" value function given by

$$V_t(\mathbf{x}_t) = \log \int \exp\left(Q_t(\mathbf{x}_t, \mathbf{u}_t)\right) d\mathbf{u}_t.$$

Under linear dynamics and quadratic costs, $V$ has the same form as in the LQR derivation above, which means that $\pi_{\mathcal{G}}(\mathbf{u}_t | \mathbf{x}_t)$ is a linear Gaussian with mean $g(\mathbf{x}_t)$ and covariance $Q_{\mathbf{uu}t}^{-1}$ [10]. Together with the linearized dynamics, the resulting policy specifies a Gaussian distribution over trajectories with Markovian independence:

$$q(\zeta) = \tilde{p}(\mathbf{x}_t) \prod_{t=1}^{T} \pi_{\mathcal{G}}(\mathbf{u}_t | \mathbf{x}_t) \tilde{p}(\mathbf{x}_{t+1} | \mathbf{x}_t, \mathbf{u}_t),$$

where $\pi_{\mathcal{G}}(\mathbf{u}_t | \mathbf{x}_t) = \mathcal{N}(g(\mathbf{x}_t), Q_{\mathbf{uu}t}^{-1})$, $\tilde{p}(\mathbf{x}_t)$ is an initial state distribution, and $\tilde{p}(\mathbf{x}_{t+1} | \mathbf{x}_t, \mathbf{u}_t) = \mathcal{N}(f_{\mathbf{x}t}\hat{\mathbf{x}}_t + f_{\mathbf{u}t}\hat{\mathbf{u}}_t + \bar{\mathbf{x}}_{t+1}, \Sigma_{ft})$ is the linearized dynamics with Gaussian noise $\Sigma_{ft}$. This distribution also corresponds to a Laplace approximation for $p(\zeta | \mathcal{O}, \theta)$, which is formed from the exponential of the second order Taylor expansion of $\log p(\zeta | \mathcal{O}, \theta)$ [15].

Once we compute $\pi_{\mathcal{G}}(\mathbf{u}_t | \mathbf{x}_t)$ using iterative LQR/DDP, it is straightforward to obtain the marginal distributions $q(\mathbf{x}_t)$, which will be useful in the next section for minimizing the variational bound $\mathcal{L}(q, \theta)$. Using $\mu_t$ and $\Sigma_t$ to denote the mean and covariance of the marginal at time $t$ and assuming that the initial state distribution at $t = 1$ is given, the marginals can be computed recursively as

$$\mu_{t+1} = \begin{bmatrix} f_{\mathbf{x}t} & f_{\mathbf{u}t} \end{bmatrix} \begin{bmatrix} \mu_t \\ \bar{\mathbf{u}}_t + \mathbf{k}_t + \mathbf{K}_t(\mu_t - \bar{\mathbf{x}}_t) \end{bmatrix}$$

$$\Sigma_{t+1} = \begin{bmatrix} f_{\mathbf{x}t} & f_{\mathbf{u}t} \end{bmatrix} \begin{bmatrix} \Sigma_t & \Sigma_t \mathbf{K}_t^{\mathrm{T}} \\ \mathbf{K}_t \Sigma_t & Q_{\mathbf{uu}t}^{-1} + \mathbf{K}_t \Sigma_t \mathbf{K}_t^{\mathrm{T}} \end{bmatrix} \begin{bmatrix} f_{\mathbf{x}t} & f_{\mathbf{u}t} \end{bmatrix}^{\mathrm{T}} + \Sigma_{ft}.$$

**Algorithm 1** Variational Guided Policy Search
---
1: Initialize $q(\zeta)$ using DDP with cost $\bar{c}(\mathbf{x}_t, \mathbf{u}_t) = \alpha_0 c(\mathbf{x}_t, \mathbf{u}_t)$
2: **for** iteration $k = 1$ to $K$ **do**
3:     Compute marginals $(\mu_1, \Sigma_t), \ldots, (\mu_T, \Sigma_T)$ for $q(\zeta)$
4:     Optimize $\mathcal{L}(q, \theta)$ with respect to $\theta$ using standard nonlinear optimization methods
5:     Set $\alpha_k$ based on annealing schedule, for example $\alpha_k = \exp\left(\frac{K-k}{K}\log\alpha_0 + \frac{k}{K}\log\alpha_K\right)$
6:     Optimize $q(\zeta)$ using DDP with cost $\bar{c}(\mathbf{x}_t, \mathbf{u}_t) = \alpha_k c(\mathbf{x}_t, \mathbf{u}_t) - \log\pi_\theta(\mathbf{u}_t|\mathbf{x}_t)$
7: **end for**
8: Return optimized policy $\pi_\theta(\mathbf{u}_t|\mathbf{x}_t)$
---

When the dynamics are nonlinear or the modified cost $\bar{c}(\mathbf{x}_t, \mathbf{u}_t)$ is nonquadratic, this solution only approximates the minimum of the KL divergence. In practice, the approximation is quite good when the dynamics and the cost $c(\mathbf{x}_t, \mathbf{u}_t)$ are smooth. Unfortunately, the policy term $\log\pi_\theta(\mathbf{u}_t|\mathbf{x}_t)$ in the modified cost $\bar{c}(\mathbf{x}_t, \mathbf{u}_t)$ can be quite jagged early on in the optimization, particularly for nonlinear policies. To mitigate this issue, we compute the derivatives of the policy not only along the current trajectory, but also at samples drawn from the current marginals $q(\mathbf{x}_t)$, and average them together. This averages out local perturbations in $\log\pi_\theta(\mathbf{u}_t|\mathbf{x}_t)$ and improves the approximation. In Section 8, we discuss more sophisticated techniques that could be used in future work to handle highly nonlinear dynamics for which this approximation may be inadequate.

## 5 Variational Guided Policy Search

The variational guided policy search (variational GPS) algorithm alternates between minimizing the KL divergence in Equation 2 with respect to $q(\zeta)$ as described in the previous section, and maximizing the bound $\mathcal{L}(q, \theta)$ with respect to the policy parameters $\theta$. Minimizing the KL divergence reduces the difference between $\mathcal{L}(q, \theta)$ and $\log p(\mathcal{O}|\theta)$, so that the maximization of $\mathcal{L}(q, \theta)$ becomes a progressively better approximation for the maximization of $\log p(\mathcal{O}|\theta)$. The method is summarized in Algorithm 1. The bound $\mathcal{L}(q, \theta)$ can be maximized by a variety of standard optimization methods, such as stochastic gradient descent (SGD) or LBFGS. The gradient is given by

$$\nabla\mathcal{L}(q, \theta) = \int q(\zeta) \sum_{t=1}^{T} \nabla\log\pi_\theta(\mathbf{u}_t|\mathbf{x}_t)d\zeta \approx \frac{1}{M}\sum_{i=1}^{M}\sum_{t=1}^{T}\nabla\log\pi_\theta(\mathbf{u}_t^i|\mathbf{x}_t^i), \qquad (3)$$

where the samples $(\mathbf{x}_t^i, \mathbf{u}_t^i)$ are drawn from the marginals $q(\mathbf{x}_t, \mathbf{u}_t)$. When using SGD, new samples can be drawn at every iteration, since sampling from $q(\mathbf{x}_t, \mathbf{u}_t)$ only requires the precomputed marginals from the preceding section. Because the marginals are computed using linearized dynamics, we can be assured that the samples will not deviate drastically from the optimized trajectory, regardless of the true dynamics. The resulting SGD optimization is analogous to a supervised learning task with an infinite training set. When using LBFGS, a new sample set can generated every $n$ LBFGS iterations. We found that values of $n$ from 20 to 50 produced good results.

When choosing the policy class, it is common to use deterministic policies with additive Gaussian noise. In this case, we can optimize the policy more quickly and with many fewer samples by only sampling states and evaluating the integral over actions analytically. Letting $\mu_{\mathbf{x}_t}^\theta, \Sigma_{\mathbf{x}_t}^\theta$ and $\mu_{\mathbf{x}_t}^q, \Sigma_{\mathbf{x}_t}^q$ denote the means and covariances of $\pi_\theta(\mathbf{u}_t|\mathbf{x}_t)$ and $q(\mathbf{u}_t|\mathbf{x}_t)$, we can write $\mathcal{L}(q, \theta)$ as

$$\mathcal{L}(q, \theta) \approx \frac{1}{M}\sum_{i=1}^{M}\sum_{t=1}^{T}\int q(\mathbf{u}_t|\mathbf{x}_t^i)\log\pi_\theta(\mathbf{u}_t|\mathbf{x}_t^i)d\mathbf{u}_t + \text{const}$$

$$= \frac{1}{M}\sum_{i=1}^{M}\sum_{t=1}^{T} -\frac{1}{2}\left(\mu_{\mathbf{x}_t^i}^\theta - \mu_{\mathbf{x}_t^i}^q\right)^{\mathrm{T}}\Sigma_{\mathbf{x}_t^i}^{\theta-1}\left(\mu_{\mathbf{x}_t^i}^\theta - \mu_{\mathbf{x}_t^i}^q\right) - \frac{1}{2}\log\left|\Sigma_{\mathbf{x}_t^i}^\theta\right| - \frac{1}{2}\text{tr}\left(\Sigma_{\mathbf{x}_t^i}^{\theta-1}\Sigma_{\mathbf{x}_t^i}^q\right) + \text{const}.$$

Two additional details should be taken into account in order to obtain the best results. First, although model-based trajectory optimization is more powerful than random exploration, complex tasks such as bipedal locomotion, which we address in the following section, are too difficult to solve entirely with trajectory optimization. To solve such tasks, we can initialize the procedure from a good initial

trajectory, typically provided by a demonstration. This trajectory is only used for initialization and need not be reproducible by any policy, since it will be modified by subsequent DDP invocations.

Second, unlike the average cost objective, the maximum likelihood objective is sensitive to the magnitude of the cost. Specifically, the logarithm of Equation 1 corresponds to a soft minimum over all likely trajectories under the current policy, with the softness of the minimum inversely proportional to the cost magnitude. As the magnitude increases, this objective scores policies based primarily on their best-case cost, rather than the average case. As the magnitude decreases, the objective becomes more similar to the classic average cost. Because of this, we found it beneficial to gradually anneal the cost by multiplying it by $\alpha_k$ at the $k^{\text{th}}$ iteration, starting with a high magnitude to favor aggressive exploration, and ending with a low magnitude to optimize average case performance. In our experiments, $\alpha_k$ begins at 1 and is reduced exponentially to 0.1 by the $50^{\text{th}}$ iteration.

Since our method produces both a parameterized policy $\pi_\theta(\mathbf{u}_t|\mathbf{x}_t)$ and a DDP solution $\pi_{\mathcal{G}}(\mathbf{u}_t|\mathbf{x}_t)$, one might wonder why the DDP policy itself is not a suitable controller. The issue is that $\pi_\theta(\mathbf{u}_t|\mathbf{x}_t)$ can have an arbitrary parameterization, and admits constraints on available information, stationarity, etc., while $\pi_{\mathcal{G}}(\mathbf{u}_t|\mathbf{x}_t)$ is always a nonstationary linear feedback policy. This has three major advantages: first, only the learned policy may be usable at runtime if the information available at runtime differs from the information during training, for example if the policy is trained in simulation and executed on a physical system with limited sensors. Second, if the policy class is chosen carefully, we might hope that the learned policy would generalize better than the DDP solution, as shown in previous work [10]. Third, multiple trajectories can be used to train a single policy from different initial states, creating a single controller that can succeed in a variety of situations.

## 6 Experimental Evaluation

We evaluated our method on two simulated planar locomotion tasks: swimming and bipedal walking. For both tasks, the policy sets joint torques on a simulated robot consisting of rigid links. The swimmer has 3 links and 5 degrees of freedom, including the root position, and a 10-dimensional state space that includes joint velocities. The walker has 7 links, 9 degrees of freedom, and 18 state dimensions. Due to the high dimensionality and nonlinear dynamics, these tasks represent a significant challenge for direct policy learning. The cost function for the walker was given by

$$c(\mathbf{x}, \mathbf{u}) = w_{\mathbf{u}}\|\mathbf{u}\|^2 + (v_x - v_x^\star)^2 + (p_y - p_y^\star)^2,$$

where $v_x$ and $v_x^\star$ are the current and desired horizontal velocities, $p_y$ and $p_y^\star$ are the current and desired heights of the hips, and the torque penalty was set to $w_{\mathbf{u}} = 10^{-4}$. The swimmer cost excludes the height term and uses a lower torque penalty of $w_{\mathbf{u}} = 10^{-5}$. As discussed in the previous section, the magnitude of the cost was decreased by a factor of 10 during the first 50 iterations, and then remained fixed. Following previous work [10], the trajectory for the walker was initialized with a demonstration from a hand-crafted locomotion system [22].

The policy was represented by a neural network with one hidden layer and a soft rectifying nonlinearity of the form $a = \log(1 + \exp(z))$, with Gaussian noise at the output. Both the weights of the neural network and the diagonal covariance of the output noise were learned as part of the policy optimization. The number of policy parameters ranged from 63 for the 5-unit swimmer to 246 for the 10-unit walker. Due to its complexity and nonlinearity, this policy class presents a challenge to traditional policy search algorithms, which often focus on compact, linear policies [8].

Figure 1 shows the average cost of the learned policies on each task, along with visualizations of the swimmer and walker. Methods that sample from the current policy use 10 samples per iteration, unless noted otherwise. To ensure a fair comparison, the vertical axis shows the average cost $E[c(\zeta)]$ rather than the maximum likelihood objective $\log p(\mathcal{O}|\theta)$. The cost was evaluated for both the actual stochastic policy (solid line), and a deterministic policy obtained by setting the variance of the Gaussian noise to zero (dashed line). Each plot also shows the cost of the initial DDP solution. Policies with costs significantly above this amount do not succeed at the task, either falling in the case of the walker, or failing to make forward progress in the case of the swimmer. Our method learned successful policies for each task, and often converged faster than previous methods, though performance during early iterations was often poor. We believe this is because the variational bound $\mathcal{L}(q, \theta)$ does not become a good proxy for $\log p(\mathcal{O}|\theta)$ until after several invocations of DDP, at which point the algorithm is able to rapidly improve the policy.

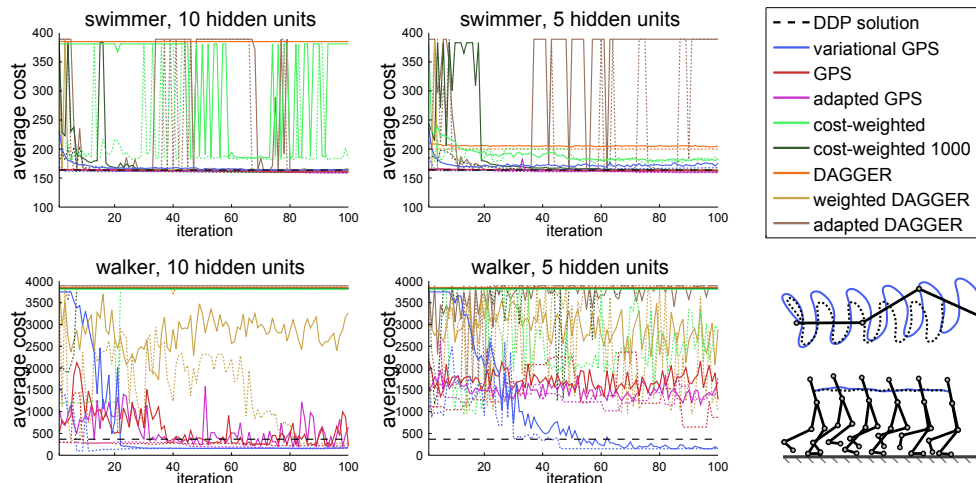

Figure 1: Comparison of variational guided policy search (VGPS) with prior methods. The average cost of the stochastic policy is shown with a solid line, and the average cost of the deterministic policy without Gaussian noise is shown with a dashed line. The bottom-right panel shows plots of the swimmer and walker, with the center of mass trajectory under the learned policy shown in blue, and the initial DDP solution shown in black.

The first method we compare to is guided policy search (GPS), which uses importance sampling to introduce samples from the DDP solution into a likelihood ratio policy search [10]. The GPS algorithm first draws a fixed number of samples from the DDP solution, and then adds on-policy samples at each iteration. Like our method, GPS uses DDP to explore regions of low cost, but the policy optimization is done using importance sampling, which can be susceptible to degenerate weights in high dimensions. Since standard GPS only samples from the initial DDP solution, these samples are only useful if they can be reproduced by the policy class. Otherwise, GPS must rely on random exploration to improve the solution. On the easier swimmer task, the GPS policy can reproduce the initial trajectory and succeeds immediately. However, GPS is unable to find a successful walking policy with only 5 hidden units, which requires modifications to the initial trajectory. In addition, although the deterministic GPS policy performs well on the walker with 10 hidden units, the stochastic policy fails more often. This suggests that the GPS optimization is not learning a good variance for the Gaussian policy, possibly because the normalized importance sampled estimator places greater emphasis on the relative probability of the samples than their absolute probability.

The adaptive variant of GPS runs DDP at every iteration and adapts to the current policy, in the same manner as our method. However, samples from this adapted DDP solution are then included in the policy optimization with importance sampling, while our approach optimizes the variational bound $\mathcal{L}(q, \theta)$. In the GPS estimator, each sample $\zeta_i$ is weighted by an importance weight dependent on $\pi_\theta(\zeta_i)$, while the samples in our optimization are not weighted. When a sample has a low probability under the current policy, it is ignored by the importance sampled optimizer. Because of this, although the adaptive variant of GPS improves on the standard variant, it is still unable to learn a walking policy with 5 hidden units, while our method quickly discovers an effective policy.

We also compared to an imitation learning method called DAGGER. DAGGER aims to learn a policy that imitates an oracle [14], which in our case is the DDP solution. At each iteration, DAGGER adds samples from the current policy to a dataset, and then optimizes the policy to take the oracle action at each dataset state. While adjusting the current policy to match the DDP solution may appear similar to our approach, we found that DAGGER performed poorly on these tasks, since the on-policy samples initially visited states that were very far from the DDP solution, and therefore the DDP action at these states was large and highly suboptimal. To reduce the impact of these poor states, we implemented a variant of DAGGER which weighted the samples by their probability under the DDP marginals. This variant succeeded on the swimming tasks and eventually found a good deterministic policy for the walker with 10 hidden units, though the learned stochastic policy performed very poorly. We also implemented an adapted variant, where the DDP solution is reoptimized at each iteration to match the policy (in addition to weighting), but this variant performed

worse. Unlike DAGGER, our method samples from a Gaussian distribution around the current DDP solution, ensuring that all samples are drawn from good parts of the state space. Because of this, our method is much less sensitive to poor or unstable initial policies.

Finally, we compare to an alternative variational policy search algorithm analogous to PoWER [8]. Although PoWER requires a linear policy parameterization and a specific exploration strategy, we can construct an analogous non-linear algorithm by replacing the analytic M-step with nonlinear optimization, as in our method. This algorithm is identical to ours, except that instead of using DDP to optimize $q(\zeta)$, the variational distribution is formed by taking samples from the current policy and reweighting them by the exponential of their cost. We call this method "cost-weighted." The policy is still initialized with supervised training to resemble the initial DDP solution, but otherwise this method does not benefit from trajectory optimization and relies entirely on random exploration. This kind of exploration is generally inadequate for such complex tasks. Even if the number of samples per iteration is increased to $10^3$ (denoted as "cost-weighted 1000"), this method still fails to solve the harder walking task, suggesting that simply taking more random samples is not the solution.

These results show that our algorithm outperforms prior methods because of two advantages: we use a model-based trajectory optimization algorithm instead of random exploration, which allows us to outperform model-free methods such as the "cost-weighted" PoWER analog, and we decompose the policy search into two simple optimization problems that can each be solved efficiently by standard algorithms, which leaves us less vulnerable to local optima than more complex methods like GPS.

# 7   Previous Work

In optimizing a maximum likelihood objective, our method builds on previous work that frames control as inference [20, 19, 13]. Such methods often redefine optimality in terms of a log evidence probability, as in Equation 1. Although this definition differs from the classical expected return, our evaluation suggests that policies optimized with respect to this measure also exhibit a good average return. As we discuss in Section 5, this objective is risk seeking when the cost magnitude is high, and annealing can be used to gradually transition from an objective that favors aggressive exploration to one that resembles the average return. Other authors have also proposed alternative definitions of optimality that include appealing properties like maximization of entropy [23] or computational benefits [16]. However, our work is the first to our knowledge to show how trajectory optimization can be used to guide policy learning within the control-as-inference framework.

Our variational decomposition follows prior work on policy search with variational inference [3, 11] and expectation maximization [8, 21]. Unlike these methods, our approach aims to find a variational distribution $q(\zeta)$ that is best suited for control and leverages a known dynamics model. We present an interpretation of the KL divergence minimization in Equation 2 as model-based exploration, which can be performed with a variant of DDP. As shown in our evaluation, this provides our method with a significant advantage over methods that rely on model-free random exploration, though at the cost of requiring a differentiable model of the dynamics. Interestingly, our algorithm never requires samples to be drawn from the current policy. This can be an advantage in applications where running an unstable, incompletely optimized policy can be costly or dangerous.

Our use of DDP to guide the policy search parallels our previous Guided Policy Search (GPS) algorithm [10]. Unlike the proposed method, GPS incorporates samples from DDP directly into an importance-sampled estimator of the return. These samples are therefore only useful when the policy class can reproduce them effectively. As shown in the evaluation of the walker with 5 hidden units, GPS may be unable to discover a good policy when the policy class cannot reproduce the initial DDP solution. Adaptive GPS addresses this issue by reoptimizing the trajectory to resemble the current policy, but the policy is still optimized with respect to an importance-sampled return estimate, which leaves it highly prone to local optima, and the theoretical justification for adaptation is unclear. The proposed method justifies the reoptimization of the trajectory under a variational framework, and uses standard maximum likelihood in place of the complex importance-sampled objective.

We also compared our method to DAGGER [14], which uses a general-purpose supervised training algorithm to train the current policy to match an oracle, which in our case is the DDP solution. DAGGER matches actions from the oracle policy at states visited by the current policy, under the

assumption that the oracle can provide good actions in all states. This assumption does not hold for DDP, which is only valid in a narrow region around the trajectory. To mitigate the locality of the DDP solution, we weighted the samples by their probability under the DDP marginals, which allowed DAGGER to solve the swimming task, but it was still outperformed by our method on the walking task, even with adaptation of the DDP solution. Unlike DAGGER, our approach is relatively insensitive to the instability of the learned policy, since the learned policy is not sampled.

Several prior methods also propose to improve policy search by using a distribution over high-value states, which might come from a DDP solution [6, 1]. Such methods generally use this "restart" distribution as a new initial state distribution, and show that optimizing a policy from such a restart distribution also optimizes the expected return. Unlike our approach, such methods only use the states from the DDP solution, not the actions, and tend to suffer from the increased variance of the restart distribution, as shown in previous work [10].

## 8 Discussion and Future Work

We presented a policy search algorithm that employs a variational decomposition of a maximum likelihood objective to combine trajectory optimization with policy search. The variational distribution is obtained using differential dynamic programming (DDP), and the policy can be optimized with a standard nonlinear optimization algorithm. Model-based trajectory optimization effectively takes the place of random exploration, providing a much more effective means for finding low cost regions that the policy is then trained to visit. Our evaluation shows that this algorithm outperforms prior variational methods and prior methods that use trajectory optimization to guide policy search.

Our algorithm has several interesting properties that distinguish it from prior methods. First, the policy search does not need to sample the learned policy. This may be useful in real-world applications where poor policies might be too risky to run on a physical system. More generally, this property improves the robustness of our method in the face of unstable initial policies, where on-policy samples have extremely high variance. By sampling directly from the Gaussian marginals of the DDP-induced distribution over trajectories, our approach also avoids some of the issues associated with unstable dynamics, requiring only that the task permit effective trajectory optimization.

By optimizing a maximum likelihood objective, our method favors policies with good best-case performance. Obtaining good best-case performance is often the hardest part of policy search, since a policy that achieves good results occasionally is easier to improve with standard on-policy search methods than one that fails outright. However, modifying the algorithm to optimize the standard average cost criterion could produce more robust controllers in the future.

The use of local linearization in DDP results in only approximate minimization of the KL divergence in Equation 2 in nonlinear domains or with nonquadratic policies. While we mitigate this by averaging the policy derivatives over multiple samples from the DDP marginals, this approach could still break down in the presence of highly nonsmooth dynamics or policies. An interesting avenue for future work is to extend the trajectory optimization method to nonsmooth domains by using samples rather than linearization, perhaps analogously to the unscented Kalman filter [5, 18]. This could also avoid the need to differentiate the policy with respect to the inputs, allowing for richer policy classes to be used. Another interesting avenue for future work is to apply model-free trajectory optimization techniques [7], which would avoid the need for a model of the system dynamics, or to learn the dynamics from data, for example by using Gaussian processes [2]. It would also be straightforward to use multiple trajectories optimized from different initial states to learn a single policy that is able to succeed under a variety of initial conditions.

Overall, we believe that trajectory optimization is a very useful tool for policy search. By separating the policy optimization and exploration problems into two separate phases, we can employ simpler algorithms such as SGD and DDP that are better suited for each phase, and can achieve superior performance on complex tasks. We believe that additional research into augmenting policy learning with trajectory optimization can further advance the performance of policy search techniques.

**Acknowledgments**

We thank Emanuel Todorov, Tom Erez, and Yuval Tassa for providing the simulator used in our experiments. Sergey Levine was supported by NSF Graduate Research Fellowship DGE-0645962.

## Footnotes

[1] The minimization is not exact if the dynamics $p(\mathbf{x}_{t+1} | \mathbf{x}_t, \mathbf{u}_t)$ are not deterministic, but the result is very close if the dynamics have much lower entropy than the policy and exponentiated cost, which is often the case.

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
