[Reviews · NeurIPS 2013]

Submitted by Assigned_Reviewer_2

The paper under review, "Variational Guided Policy Search" introduces a new approach of how classical policy search can be combined and improved with trajectory optimization methods serving as exploration strategy. An optimization criteria with the goal of finding optimal policy parameters is decomposed with a variational approach. The variational distribution is approximated as Gaussian distribution which allows a solution with the iterative LQR algorithm. The overall algorithm uses expectation maximization to iterate between minimizing the KL divergence of the variational decomposition and maximizing the lower bound with respect to the policy parameters.

The paper has a high quality with a sound mathematical formulation and a step by step derivation of the algorithm. The authors describe their approach and the used previous work very clearly. Additionally, they also provide a nice overview about previous work in this area.
The paper is original in the sense of using a variational framework for combining policy search with trajectory optimization. The major advantage over other policy search methods is that no samples from the current policy are required. This is useful in real-world applications where it is risky to execute unstable policies. The experiments show that the presented algorithm is capable of learning tasks with a reasonable amount of iterations. Especially in real-world applications (e.g., robots) where exploration on the real system is dangerous, this approach might be an alternative to current state of the art policy search algorithms.

Some minor comments:
- Provide a comparison of computational costs in the experiments
- Add the number of learned parameters in the experiments
- line 251: provide the equation of the neural network policy
Summary: The paper "Variational Guided Policy Search" provides a very strong contribution to the field of policy search algorithms. The mathematical formulation combining policy search with trajectory optimization opens new directions for future research and applications in this area.

Submitted by Assigned_Reviewer_6

The paper describes a method called variational guided policy search (VGPS) that alternates policy optimization and trajectory optimization. Exploration is here done by trajectory optimization.

The quality of the paper is exceptional.
It is also clearly written and has a high significance.
The theoretical foundation is to my understanding flawless. The experiments are done in simulation only, but are adequately complex. The presented method performs better and more robust than comparable methods. Especially impressive is the improvement in performance and robustness compared to the PoWER variant (cost-weighted).

To my understanding the paper needs no improvement, I didn't even find any typos - so I suggest acceptance.
Summary: The paper describes a method called variational guided policy search (VGPS) that alternates policy optimization and trajectory optimization (for exploration and guiding the policy search). I found nothing that is worth mentioning as criticism. The method is well founded theoretically and performs better than standard policy search methods on the chosen complex locomotion tasks.

Submitted by Assigned_Reviewer_7

Authors describe a model based policy search method that alternates between policy optimization and trajectory optimization. This split in policy optimization is obtained using variational methods. The optimization is then performed using simpler algorithms like SGD and DDP. The application section then describes two simulated experiments using the algorithm.
The idea is an improvement on previously described Guided Policy search algorithm. Using variational methods to split the policy search is a novel idea, and it is described quite clearly in the paper, but I do have a few concerns about the paper:
1) The problem seems easier solved using trajectory optimization with SGD, but how is the problem of choosing a learning rate decided. The results show a slower convergence compared to GPS (the previous method) that is attributed to a slower convergence of the trajectory optimization. Overall I would like a clarification if the learning rates can solve this problem.
2) There needs to be an explanation towards why in the swimming simulation GPS performs marginally better than variational GPS, even with fewer parameters.
3) PoWER is a model free approach using EM, and it specifically avoids gradient based methods. How does it compare when it is implemented with a gradient based solution? Moreover, this explanation seemed hurried.
The idea is innovative, but I would like more clarity from the results section.
Summary: Overall the idea is innovative, but I would like more clarity from the results section.

Submitted by Assigned_Reviewer_8

The paper presents a new model-based policy search algorithm that is based on the variational decomposition of policy search. I order to estimate the variational distribution, a linearized model is used to minimize the KL-term involved in determining the variational distribution. The resulting algorithm outperforms existing methods which is shown on challenging benchmark tasks.

I personally think the approach might be promising and i also like the experiments, but I think there are crucial comparisons missing. First of all, the presented method is a model-based method, and, hence the authors should also compare against other model-based approaches. For example, the pilco framework [1] learns a GP model for long-term prediction with moment matching (which is actually a very similar approximation as the one performed by ddp). The policy is then updated by Bfgs. This algorithm is state of the art in model-based policy search and it is not clear to me why the presented algorithm should perform better than pilco. Furthermore, the authors claim that the ddp solution has several disadvantages to the learned policy. While this might be well the case, the authors should also show that in their evaluations. Maybe the generalization of the learned policy is better but the ddp solution has higher quality for a single task. That would be good to know. The comparison to the variant of power is also rather unfair as power does not use a model. However, as the model is assumed to be known, the model can be used to generate a vast amount of samples. In the limit of an infinite amount of samples, power should produce the same results as the ddp solution?. It would be interesting to know how many samples are needed to match the performance of the ddp solution.
In theory, the presented method probably simplifies the computational demands in comparison to the samplebased version of power, but, in the limit of infinite samples, the result should be the same. Hence, the authors need to evaluate the computational benefits of the presented method. Furthermore, the relationship to existing state of the art stochastic optimal control (soc) algorithms, such as soc by approximate inference [2] or policy improvements by path integrals [3] should at least be discussed.

Minor issues: - the choice of alpha_k seems to be very hacky to me. See [4] for a more sophisticated approach for choosing the temperature of the exponential transformation. - I do not really get why the exploration strategy of power is "random" while the exploration strategy of ddp is guided. Both use exactly the same exploration policy. While power uses samples to obtain the variational distribution, ddp uses approximation to determine a closed form solution for q. In the limit of infinite samples, power should even produce the better results as no approximations are used. - how does your method perform if the model is not known, but needs to be learned from data, e.g., by using a Gaussian Process?

[1] Deisenroth et. al. : PILCO: A model-based and data-efficient approach to policy search [2] Rawik et. al.: Stochastic optimal control by approximate inference [3] Theoudorou et. al.: A Generalized Path Integral Control Approach to Reinforcement Learning [4] Kupscik et. al.: Data-Efficient Generalization of Robot Skills with Contextual Policy Search
Summary: While the approach could be promising, important evaluations and comparisons are missing to evaluate the contribution of the paper.

In the rebuttal the authors could invalidate my main critic-points. I think the paper can be published.
Author Feedback

Author rebuttal: We thank the reviewers for their assessment. We believe that interleaving policy learning and trajectory optimization is a promising direction for improving policy search algorithms, and hope that this paper can be a step in that direction.

Reviewer #8 suggests PILCO as a comparison. PILCO uses GPs to learn an unknown model. We assume a known model, so the methods are orthogonal and complementary. We were unable to compare to PILCO directly, since it requires policies where the state can be marginalized out analytically. Our neural network policies are too complex for this. We did run PILCO with RBF policies, but were unable to produce successful locomotion behaviors. It may be that more parameter tuning is required, but the strong smoothness assumptions of the PILCO GP model are known to fare poorly in contact-rich tasks such as walking.

Regarding the PoWER-like cost-weighted method, we agree with #8 and #7 that the comparison is unfair, since PoWER is model-free. PoWER was chosen for its structural similarity (it uses EM), while GPS was chosen as the main competitor, since it also uses a model. We disagree with #8's comment that a large number of samples would permit PoWER to solve the task. While in the limit of infinite computation this is true, such cases can be solved by any other model-free method, such as REINFORCE. In practice, infinite samples are impossible even with a known model, so sample complexity matters. In the worst case, only a few of the possible trajectories are good, and the number of samples can scale exponentially in the dimensionality of the trajectory (N*T). Algorithms that scale exponentially are not merely expensive, they are intractable. Experimentally, we found that the "cost-weighted" method could not solve the walking task with either 1000 or 10000 samples per iteration (we tried 100000 but ran out of RAM).

#8 also notes other RL methods by Theodorou and Rawlik. Like the cost-weighted method, they use model-free policy search. We would be happy to discuss them, but we do not believe their performance on our tasks would differ much from "cost-weighted," since they too rely on random exploration. We disagree with the reviewer that the random exploration strategy is identical to DDP, since DDP uses the gradient of the dynamics to improve a trajectory, while PoWER relies on randomly sampling good trajectories. If the current policy is uninformative (as it is in the beginning), a good trajectory must be found "accidentally." In walking for example, this is profoundly unlikely, which is why random exploration methods fare poorly in this experiment.

#8 also brings up generalization. While we do not explore how well neural network policies generalize beyond DDP, this was evaluated in detail in "Guided Policy Search" (Levine et al. '13), using very similar tasks and a very similar policy class. More generally, stationary policies are often preferred for periodic tasks such as locomotion over the time-varying policies produced by DDP.

Reviewer #7 points out that GPS sometimes converges faster than VGPS. The slower convergence of VGPS can be attributed to "disagreement" between the DDP solution and the policy, which causes the policy to adopt a wider variance until they agree, at which point convergence is fast. We would be happy to further analyze the influence of learning rate on convergence in the final version. That said, VGPS is still able to solve tasks that GPS cannot, such as the 5-hidden-unit walker. In regard to SGD learning rate, we used the same rate in all tests. Using improvements like AdaGrad makes the choice of learning rate less critical. We also have a version of VGPS that uses LBFGS and produces very similar results, though SGD is known to scale better.

#7 mentions a gradient-based alternative to PoWER. We tested standard policy gradient algorithms, but found that they did not perform better than PoWER. For a model-based comparison, we included GPS.